# RANDOM WALK DIFFUSION FOR GRAPH GENERATION

## ABSTRACT

Graph generation addresses the problem of generating new graphs that have a data
distribution similar to real-world graphs. Recently, the task of graph generation
has gained increasing attention with applications ranging from data augmenta-
tion to constructing molecular graphs with specific properties. Previous diffusion-
based approaches have shown promising results in terms of the quality of the gen-
erated graphs. However, most methods are designed for generating small graphs
and do not scale well to large graphs. In this work, we introduce ARROW-Diff, a
novel random walk-based diffusion approach for graph generation. It utilizes an
order agnostic autoregressive diffusion model enabling us to generate graphs at a
very large scale. ARROW-Diff encompasses an iterative procedure that builds the
final graph from sampled random walks based on an edge classification task and
directed by node degrees. Our method outperforms all baseline methods in terms
of training and generation time and can be trained both on single- and multi-graph
datasets. Moreover, it outperforms most baselines on multiple graph statistics re-
flecting the high quality of the generated graphs.

## 1 INTRODUCTION

Graph generation addresses the problem of generating graphs similar to real-world ones, with ap-
plications ranging from modeling social interactions to constructing knowledge graphs, as well as
designing new molecular structures. Traditional methods for graph generation focused on generat-
ing graphs with a predefined characteristic (Erdős et al., 1960; Barabási & Albert, 1999). Because
of their handed-crafted nature, these methods fail to capture other graph properties as in the work of
Erdős et al. (1960), where the generated graphs do not have the heavy-tailed degree distribution.

Recent deep graph generative approaches have gained increasing attention because of their ability
to learn the generative process of a graph and capture its complicated topology. Generally, these
methods comprise three building blocks (Zhu et al., 2022): (1) An encoder, which learns a dense
continuous latent representation of a graph's elements. (2) A sampler, which samples latent rep-
resentations from the learned distribution $z \sim p(z)$, and (3) a decoder, which restores the learned
latent representation into a graph structure. In the context of graph generation, the decoders can
be split into two categories: Sequential generators and one-shot generators. Sequential generation
methods include like GraphRNN (You et al., 2018), where elements of the graph, i.e., nodes or edges
are generated sequentially one-by-one or block-by-block as in Liao et al. (2019). Because of their
sequential generation process, these approaches naturally accommodate complex local dependen-
cies between the generated edges or nodes. Some important limitations of these approaches include:
(1) Their inability to account for long-term dependencies (e.g., scale-free property), and (2) the
need to implement a node ordering scheme to satisfy the permutation invariance property of graphs,
since in the general setting, a graph with $N$ nodes, can be represented with up to $N!$ equivalent
adjacency matrices. This constitutes a real challenge in larger graphs. On the other hand, one-shot
generation approaches generate a graph represented by an adjacency matrix by sampling from a
learned latent distribution in one step (Guo & Zhao, 2022). Methods that fall under this category
include GraphVAE (Simonovsky & Komodakis, 2018), VGAE (Kipf & Welling, 2016b), and Net-
GAN (Bojchevski et al., 2018). These methods generate graphs in one shot and do not require node
ordering. However, they are limited in terms of (1) scalability to larger graphs as in Simonovsky &
Komodakis (2018); Kipf & Welling (2016b), (2) the requirement for post-processing and setting a
predefined number of nodes, and (3) the independence assumption, which compromises the quality
of the generated graphs.

An even more recent body of work in generative modeling is diffusion-based probabilistic models, inspired by non-equilibrium thermodynamics and first introduced by Sohl-Dickstein et al. (2015). Since then, this class of generative models has been applied in various domains including image and video, outperforming all state-of-the-art methods (Dhariwal & Nichol, 2021; Ho et al., 2022). In short, diffusion-based generative models are parameterized Markov chains that learn the generative process by modeling the reverse of a diffusion process, which gradually corrupts the input data $x$ until it reaches pure noise. Diffusion-based methods for graph generation can be divided into two main categories. The first one includes methods that implement diffusion in the continuous space e.g., by adding Gaussian noise to the node features and graph adjacency matrix (Niu et al., 2020; Jo et al., 2022). This form of diffusion however makes it difficult to capture the underlying structure of graphs since it destroys the sparsity pattern of graphs (Vignac et al., 2023). The second one includes methods that are based on diffusion in the discrete space (Vignac et al., 2023; Haefeli et al., 2022; Chen et al., 2023) by successive graph edits e.g., additions or deletions of edges/nodes or edge/node features. Diffusion-based graph generation methods are invariant to node ordering and do not suffer from long-term memory dependency which makes them advantageous over (sequential) auto-regressive-based methods. However, many approaches found in the literature are only designed for small graphs (Niu et al., 2020; Jo et al., 2022; Vignac et al., 2023).

In this work, we introduce ARROW-Diff (AutoRegressive RandOm Walk Diffusion), a novel approach for graph generation based on random walk diffusion. Our work aims to scale diffusion-based models to generate very large graphs. Our contributions can be summarized as follows: (1) We introduce random walk-based diffusion using order agnostic Autoregressive Diffusion Models (OA-ARDMs) Hoogeboom et al. (2022) that enable us to learn the context of the nodes in random walks sampled from real-world graphs. (2) We propose an iterative procedure, ARROW-Diff, that builds the final graph from the sampled random walks based on an edge classification task and directed by node degrees as in Chen et al. (2023). We show that our method surpasses all baselines both in terms of the training speed of the diffusion model as well as graph generation time. Unlike most existing diffusion-based graph generation approaches, our method can scale to very large graphs such as the citation networks from McCallum et al. (2000); Sen et al. (2008); Pan et al. (2016). Moreover, our method is flexible and can be applied to learn from either a single graph or multiple input graphs.

## 2 BACKGROUND

**Discrete Diffusion Models**   Recent works show that diffusion models are applicable to discrete data (Sohl-Dickstein et al., 2015; Hoogeboom et al., 2021; Austin et al., 2021; Hoogeboom et al., 2022). The diffusion process of these models is based on the Categorical distribution over input features of a data point, instead of the Gaussian distribution. Initially, discrete diffusion models used uniform noise to corrupt the input in the forward diffusion process (Sohl-Dickstein et al., 2015; Hoogeboom et al., 2021). Later, Austin et al. (2021) extended this process and introduced a general framework for discrete diffusion (D3PM) based on Markov transition matrices $[\boldsymbol{Q}_t]_{ij} = q(x_t = j|x_{t-1} = i)$ for categorical random variables $x_{t-1}, x_t \in \{1, 2, \ldots, K\}$. One possible realization of the D3PM framework is the so-called absorbing state diffusion (Austin et al., 2021) that uses transition matrices with an additional absorbing state to stochastically mask entries of data points in each forward diffusion step.

**Order Agnostic Autoregressive Models**   Recently, Hoogeboom et al. (2022) introduced the concept of OA-ARDMs and demonstrated the parity between autoregressive diffusion models and absorbing state diffusion (Austin et al., 2021). Unlike standard autoregressive models, order agnostic autoregressive models are able to capture dependancies in the input regardless of their temporal order. Let $x$ be a $D$-dimensional data, an Order Agnostic Autoregressive Model can generate $x$ in a random order that follows a permutation $\sigma \in S_D$, where $S_D$ denotes the set of possible permutations of $\{1, 2, \ldots, D\}$. Specifically, their log-likelihood can be written as:

$$\log p(\boldsymbol{x}) \geq \mathbb{E}_{\sigma \sim \mathcal{U}(S_D)} \sum_{t=1}^{D} \log p(x_{\sigma(t)}|\boldsymbol{x}_{\sigma(<t)}), \tag{1}$$

where $\boldsymbol{x}_{\sigma(<t)}$ represents all elements of $x$ for which $\sigma$ is less than $t$ (Hoogeboom et al., 2022). Moreover, Hoogeboom et al. (2022) show the the significant improvement in terms of training and

sampling time of OA-ARDMs in comparison to absorbing state diffusion. In this work, we use the OA-ARDM to perform diffusion on the level of random walks. The exact steps of training adapted to our case are explained in Section 4.

## 3 RELATED WORK

**One-Shot Graph Generation Models**   After the success of deep generative approaches such as Variational Autoencoders (VAEs) (Kingma & Welling, 2013) and Generative Adversarial Networks (GANs) (Goodfellow et al., 2014) in other domains, these methods have been used for graph generation. VAE-based graph generation approaches like the Variational Graph Auto-Encoder (VGAE) (Kipf & Welling, 2016b), GraphVAE (Simonovsky & Komodakis, 2018) and Graphite (Grover et al., 2019) embed a graph $G$ into a continuous latent representation $z$ using an encoder defined by a variational posterior $q_\phi(z|G)$, and a generative decoder $p_\theta(G|z)$. These models are trained by minimizing the upper bound on the negative log-likelihood $E_{q_\phi(z|G)}[-\log p_\theta(G|z)] + KL[q_\phi(z|G)||p(z)]$. However, due to their run time complexity of $\mathcal{O}(N^2)$, VAE-based graph generation approaches are unable to scale to large graphs. Bojchevski et al. (2018) presented NetGAN, a GAN-based method designed for graph generation. Specifically, it uses a generator based on a Long Short-Term Memory (LSTM) (Hochreiter & Schmidhuber, 1997) network to generate random walks. After training, the generated random walks are used to construct a score matrix from which the edges of the generated graph are sampled. The aforementioned approaches generate edges in an edge-independent manner, sacrificing the quality of the generated graphs and limiting their ability to reproduce some graph statistics such as triangle counts and clustering coefficient (Chanpuriya et al., 2021).

**Autoregressive Graph Generation Models**   The most scalable autoregressive methods for graph generation so far are GraphRNN (You et al., 2018) and GRAN (Liao et al., 2019). These methods generate the entries of a graph adjacency matrix iteratively one entry or one block of entries at a time. To bypass the long-term bottleneck issue of RNNs, Liao et al. (2019) propose to use a Graph Neural Network (GNN) architecture instead of an RNN, which makes use of the already generated graph structure in generating the next block, and model complex dependencies between each generation step. To satisfy the permutation invariance property of graphs, these methods require a node ordering scheme. Moreover, they are only able to scale to graphs of up to 5k nodes. In the best case, the number of generation steps required for these methods is $\mathcal{O}(N)$ (Liao et al., 2019).

**Discrete Diffusion-Based Graph Generation Models**   To exploit the sparsity property of graphs, discrete diffusion-based graph generation models focus on diffusion in the discrete space i.e., on the level of the adjacency matrix (Vignac et al., 2023; Haefeli et al., 2022). In DiGress (Vignac et al., 2023), the authors propose to utilize a discrete diffusion process that diffuses on the level of categorical node and edge features. Although these approaches generate high-quality graphs (Niu et al., 2020; Jo et al., 2022) and overcome the limitation of autoregressive models, they are limited to generating very small graphs like molecules. This is because they need to make predictions for each node pair. For example, Digress has a run time complexity of $\mathcal{O}(TN^2)$, where $T$ is the number of diffusion steps and $N$ is the number of nodes, hindering it from scaling to large graphs. Currently, the only diffusion-based method that is able to scale to large graphs is EDGE (Chen et al., 2023). Here the forward process is defined by successive edge removal until an empty graph is reached. In the reverse diffusion process, only a fraction of edges are predicted based on active nodes for which the degree changes during forward diffusion. This method generates graphs with a similar degree distribution to the original graph and has a decreased run time of $\mathcal{O}(T \max(M, K^2))$, where $M$ is the number of edges in a graph and $K$ is the number of active nodes. This enables EDGE to scale to large graphs. In this work, we propose to apply the diffusion process on the level of random walks. We show that our method is therefore able to scale to very large graphs at an unprecedented size, outperforming EDGE both in terms of training and graph generation time.

## 4 GRAPH GENERATION USING RANDOM WALK DIFFUSION

In this section, we introduce ARROW-Diff, an iterative procedure that has two main components, (1) a discrete, autoregressive diffusion model that is used to sample random walks, and (2) a Graph

Neural Network (GNN) that predicts the validity of edges comprising the sampled random walks. In short, our method refines the edges of a generated graph iteratively by incorporating the edges proposed by sampled random walks into a classification task in which they are predicted either as 'valid' or as 'invalid' edges.

**Random Walk Diffusion**  Consider a graph $G = (V, E)$ with $N = |V|$ nodes. We aim to learn the (unknown) generative process $p(G)$ of $G$. Inspired by DeepWalk (Perozzi et al., 2014), node2vec (Grover & Leskovec, 2016), and by the random walk-based graph generation approach introduced by Bojchevski et al. (2018), we suggest to sample random walks from a trained diffusion model and use the edges comprising the walks as proposals for generating a new graph. To achieve this, we train an OA-ARDM (Hoogeboom et al., 2022) by viewing each node in a random walk as a word in a sentence, and follow the proposed training procedure of Hoogeboom et al. (2022) for OA-ARDMs on sequence data (Algorithm 1).

---

**Algorithm 1** Optimizing Random Walk OA-ARDMs

---

    **Input:** A random walk $\boldsymbol{x} \in V^D$, the number of nodes $N = |V|$, and a network $f$.
    **Output:** ELBO $\mathcal{L}$.
1: Sample $t \sim \mathcal{U}(1, \ldots, D)$
2: Sample $\sigma \sim \mathcal{U}(S_D)$
3: Compute $\boldsymbol{m} \leftarrow (\sigma < t)$
4: Compute $\boldsymbol{i} \leftarrow \boldsymbol{m} \odot \boldsymbol{x} + (1 - \boldsymbol{m}) \odot ((N+1) \cdot \mathbf{1}_D)$
5: $\boldsymbol{l} \leftarrow (1 - \boldsymbol{m}) \odot \log \mathcal{C}(\boldsymbol{x} | f(\boldsymbol{i}, t))$
6: $\mathcal{L}_t \leftarrow \frac{1}{D-t+1} sum(\boldsymbol{l})$
7: $\mathcal{L} \leftarrow D \cdot \mathcal{L}_t$

---

For a random walk $\boldsymbol{x} \in V^D$ of length $D$, we start by sampling a time step from a uniform distribution $t \sim \mathcal{U}(1, \ldots, D)$, and a random ordering of the nodes in the walk $\sigma \sim \mathcal{U}(S_D)$. For each time step $t$ of the diffusion process, a BERT-like (Devlin et al., 2018) training is implemented, in which $D - t + 1$ nodes (words) are masked and then predicted. To train the diffusion model, we maximize the following likelihood at each time step $t$ (Hoogeboom et al., 2022):

$$\mathcal{L}_t = \frac{1}{D-t+1} \mathbb{E}_{\sigma \sim \mathcal{U}(S_D)} \sum_{k \in \sigma(\geq t)} \log p(x_k | \boldsymbol{x}_{\sigma(<t)}) \tag{2}$$

In our case, since the input is sequence-like, the masking which is equivalent to an absorbing state (Hoogeboom et al., 2022) is done by an additional class $N + 1$. Thus, as suggested from Hoogeboom et al. (2022), the inputs to the network are (1) the masked random walk $\boldsymbol{i} = \boldsymbol{m} \odot \boldsymbol{x} + (1 - \boldsymbol{m}) \odot \boldsymbol{a}$, where $\boldsymbol{m} = \sigma < t$ is a Boolean mask, $\boldsymbol{a} = (N+1) \cdot \mathbf{1}_D$ and $\mathbf{1}_D$ is a $D$-dimensional vector of ones, and (2) the sampled time step $t$. During the training of the OA-ARDM, the random walks are sampled from the original graph.

**Conditional Random Walk Sampling**  Our ARROW-Diff graph generation approach requires the sampling of random walks starting from specific nodes. Thus, we modify the sampling procedure of Hoogeboom et al. (2022) by manually setting the first node ID of an initial random walk $\boldsymbol{x}$ to the ID of a specific node $n \in V$, i.e.

$$x_k = \begin{cases} n & \text{if } k = 1, \\ \text{mask} & \text{if } k \in \{2, \ldots, D\}. \end{cases} \tag{3}$$

Additionally, we use a restricted set of permutations $S_D^{(1)} := \{\sigma \in S_D | \sigma(1) = 1\}$, in which the order of the first element does not change after applying the permutation. To sample the remaining parts $\boldsymbol{x}_{2:D}$ of the random walk $\boldsymbol{x}$, we follow the sampling procedure of Hoogeboom et al. (2022) by starting at time step $t = 2$ and using $\sigma \sim \mathcal{U}(S_D^{(1)})$. In the following, we refer to this modified sampling of random walks as conditional random walk sampling.

**ARROW-Diff Graph Generation**  Our ARROW-Diff graph generation approach is able to generate new graphs similar to a given example using a single, original graph $G = (V, E)$. ARROW-Diff

---

**Algorithm 2** ARROW-Diff Graph Generation

    **Input:** A trained OA-ARDM, a trained GNN. The node set $V$, features $\boldsymbol{X}$ and degrees $\boldsymbol{d}_G$ of an original graph $G$ with the same node ordering as for training the OA-ARDM. The number of steps $L$ to generate the graph, the number of random walks to sample per start node $M$.

    **Output:** A generated graph $\hat{G} = (V, \hat{E})$

1: Start with an empty graph $\hat{G} = (V, \hat{E})$, where $\hat{E} = \emptyset$
2: Set the start nodes $V_{\text{start}}$ to all nodes in the graph: $V_{\text{start}} = V$
3: **for** $l = 1, \ldots, L$ **do**
4:      Sample $M$ cond. random walks for each start node $n \in V_{\text{start}}$ using the OA-ARDM: $\mathcal{R}$
5:      Compute edge proposals $\hat{E}_{\text{proposals}} := \{(n_i, n_j) \in \mathcal{R} | n_i, n_j \in V, i \neq j\}$ from $\mathcal{R}$
6:      Run the GNN on $G = (V, \hat{E} \cup \hat{E}_{\text{proposals}}, \boldsymbol{X})$ to obtain probabilities for all edges $\hat{E} \cup \hat{E}_{\text{proposals}}$
7:      Sample valid edges $\hat{E}_{\text{valid}}$ from $\hat{E} \cup \hat{E}_{\text{proposals}}$ according to the edge probabilities
8:      Edge update: $\hat{E} \leftarrow \hat{E}_{\text{valid}}$
9:      **if** $l < L$ **then**
10:        Compute the node degrees $\boldsymbol{d}_{\hat{G}}$ of $\hat{G}$ based on $\hat{E}$
11:        Compute $\boldsymbol{d} := \max(0, \boldsymbol{d}_G - \boldsymbol{d}_{\hat{G}})$
12:        Compute node-wise probabilities for each node $n \in V$: $p(n) = \frac{d_n}{\max(\boldsymbol{d})}$
13:        Sample start nodes $V_{\text{start}}$ from $V$ according to $p(n)$ using a Bernoulli distribution
14:      **end if**
15: **end for**

---

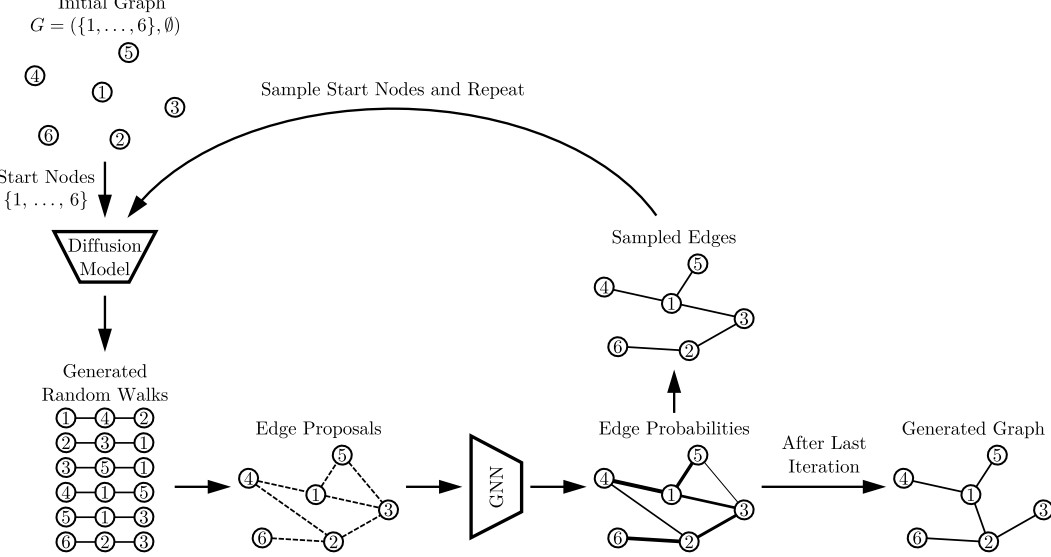

Figure 1: Overview of ARROW-Diff. Iteratively, and starting from an empty graph, a diffusion model samples conditional random walks for a set of start nodes. Then, a GNN uses the proposed edges and filters out invalid ones. This procedure is repeated using a different, sampled set of start nodes guided by the change of node degrees w.r.t. the original graph.

comprises two models: (1) An OA-ARDM (Hoogeboom et al., 2022) trained for conditional random walk sampling, and (2) a GNN trained for edge classification on perturbed versions of the original graph $G$. Specifically, the graph is corrupted by deleting edges and inserting invalid (fake) edges.

ARROW-Diff uses an iterative procedure to generate a new graph: It first starts with an empty graph $\hat{G} = (V, \emptyset)$, i.e., a graph without edges that contains the same node set $V$ as the original graph. In order to add edges to $\hat{G}$, we sample start nodes and use the trained OA-ARDM to propose new edges by sampling random walks. Similar to the work of Liao et al. (2019), we sample valid edges from the proposed ones by using the binary classification probabilities predicted by the GNN. In

Table 1: Dataset statistics of single, large-scale graph datasets used in this paper: Number of nodes, undirected edges, node features, and average node degree. For Cora-ML, Cora, CiteSeer, and DBLP, the statistics of the LCC are reported.

| Dataset | # Nodes | # Edges | # Node Features | Avg. Degree |
|---|---|---|---|---|
| Cora-ML (LCC) | 2,810 | 7,981 | 2,879 | 5.7 |
| Cora (LCC) | 18,800 | 62,685 | 8,710 | 6.7 |
| CiteSeer (LCC) | 1,681 | 2,902 | 602 | 3.5 |
| DBLP (LCC) | 16,191 | 51,913 | 1,639 | 6.4 |
| PubMed | 19,717 | 44,324 | 500 | 4.5 |

the first iteration, we use all nodes in $V$ as start nodes. In the following iterations, inspired by the degree-guided graph generation process of Chen et al. (2023), we sample start nodes from $V$ using a Bernoulli distribution by considering each node $n \in V$ according to a success probability $p(n) = \frac{d_n}{\max(\boldsymbol{d})}$, where $\boldsymbol{d} := \max(0, \boldsymbol{d}_G - \boldsymbol{d}_{\hat{G}})_n$ are the positive differences of node degrees $\boldsymbol{d}_G$ and $\boldsymbol{d}_{\hat{G}}$ from $G$ and $\hat{G}$.

The steps of our ARROW-Diff graph generation approach can be summarized in Algorithm 2 and are depicted in Figure 1. Our method is able to generate directed and undirected graphs. To generate undirected graphs, we suggest including all reverse edges to the edge proposals $\hat{E}_{\text{proposals}}$ (line 5), i.e., $(n_j, n_i) \in \mathcal{R}, n_i, n_j \in V, i \neq j$ if $(n_i, n_j) \in \hat{E}_{\text{proposals}}$, and to sample undirected edges from $\hat{E}_{\text{proposals}}$ to obtain $\hat{E}_{\text{valid}}$ (line 7).

## 5 EXPERIMENTS AND RESULTS

We split our experiments into two settings in which we train graph generation models on (1) datasets that contain only a single, large-scale graph, and (2) datasets containing multiple, small graphs. By doing so, we showcase the flexibility of our approach, ARROW-Diff, to be applied on variable size graphs, This dual experimental setting also enables us to evaluate our method against a variety of baselines which are normally optimized to either of the two settings.

### 5.1 ARROW-DIFF MODEL TRAINING AND SAMPLING

We train the OA-ARDM for random walk diffusion following the work of Hoogeboom et al. (2022), which is explained in Section 4. Specifically, we use a U-Net architecture similar to Ho et al. (2020) with one ResNet block and two levels for the down- and up-sampling processes. In the first part of our experiments, where we train on a single, large-scale graph, the walk length $D$ is set to 16 as in Bojchevski et al. (2018) and is reduced to 12 for the second setting in which we train on multiple small-scale graphs. Our iterative procedure, ARROW-Diff, is repeated for $L = 10$ times for all experiments. To generate the final graph we follow Algorithm 2, and train a 2-layer GCN (Kipf & Welling, 2016a) to classify edges into valid/invalid ones based on perturbed versions of the input graph. The full list of parameters for training the diffusion and the GNN models can be found in the supplementary materials.

### 5.2 TRAINING GRAPH GENERATION MODELS ON SINGLE-GRAPH DATASETS

**Datasets** In this setting, we use five citation graph datasets to evaluate our method: Cora-ML (McCallum et al., 2000), Cora (McCallum et al., 2000), CiteSeer (Giles et al., 1998), DBLP (Pan et al., 2016), and PubMed (Sen et al., 2008). For Cora-ML and Cora, we use the pre-processed version from Bojchevski & Günnemann (2018). Each of the five datasets contains one single, undirected, large-scale citation graph. Motivated by Bojchevski et al. (2018), we only take the largest connected component (LCC) of Cora-ML, Cora, CiteSeer, and DBLP, which all contain multiple connected components. Table 1 gives an overview of different characteristics for each graph/LCC. Similar to Bojchevski et al. (2018), we split the edge sets of each graph into training, validation, and test parts, and use only the training edges to train our model and the baseline methods.

Table 2: Graph generation results of NetGAN (Bojchevski et al., 2018), VGAE (Kipf & Welling, 2016b), Graphite (Grover et al., 2019), EDGE (Chen et al., 2023) and ARROW-Diff on the single, large-scale graph datasets from Table 1. The performance is given in terms of the mean of the edge overlap and six graph statistics across 10 generated graphs. The last column reports the graph generation time for all methods, which is the time for executing Algorithm 2 for ARROW-Diff.

| *Dataset* Methods | Max. degree | Assort- ativity | Triangle Count | Power law exp. | Avg. cl. coeff. | Global cl. coeff. | Edge Overlap | Time [s] |
|---|---|---|---|---|---|---|---|---|
| *Cora-ML* | 246 | -0.077 | 5,247 | 1.77 | 0.278 | 0.004 | - | - |
| NetGAN | 181 | -0.025 | 384 | 1.67 | 0.011 | 0.001 | 3.2% | 6.2 |
| VGAE | 948 | -0.043 | 70 M | 1.66 | 0.383 | **0.002** | 22.2% | 0.0 |
| Graphite | 115 | -0.188 | 11,532 | 1.57 | **0.201** | 0.009 | 0.3% | 0.1 |
| EDGE | **202** | **-0.051** | 1,410 | **1.76** | 0.064 | **0.002** | 1.3% | 5.5 |
| ARROW-Diff | 373 | -0.112 | **5,912** | 1.81 | 0.191 | 0.001 | **57.3%** | 1.8 |
| *Cora* | 297 | -0.049 | 48,279 | 1.69 | 0.267 | 0.007 | - | - |
| NetGAN | 135 | 0.010 | 206 | 1.61 | 0.001 | 0.000 | 0.1% | 35.0 |
| Graphite | 879 | -0.213 | 3 M | 1.31 | **0.338** | 0.001 | 0.3% | 0.9 |
| EDGE | **248** | 0.078 | 11,196 | 1.65 | 0.021 | **0.002** | 0.2% | 85.8 |
| ARROW-Diff | 536 | **-0.077** | **89,895** | **1.70** | 0.122 | **0.002** | **40.8%** | 13.7 |
| *CiteSeer* | 85 | -0.165 | 771 | 2.23 | 0.153 | 0.007 | - | - |
| NetGAN | 42 | -0.009 | 23 | 2.03 | 0.004 | 0.001 | 0.7% | 4.5 |
| VGAE | 558 | -0.036 | 15 M | 1.69 | 0.383 | 0.003 | 22.1% | 0.0 |
| Graphite | 58 | -0.198 | 2,383 | 1.70 | **0.157** | 0.016 | 0.3% | 0.1 |
| EDGE | **82** | -0.128 | 205 | 2.08 | 0.054 | 0.003 | 1.1% | 4.2 |
| ARROW-Diff | 114 | **-0.192** | **795** | **2.24** | 0.109 | **0.004** | **57.8%** | 1.6 |
| *DBLP* | 339 | -0.018 | 36,645 | 1.76 | 0.145 | 0.004 | - | - |
| NetGAN | 215 | 0.053 | 1,535 | 1.62 | 0.002 | 0.000 | 0.9% | 29.8 |
| Graphite | 734 | -0.207 | 2 M | 1.32 | 0.331 | **0.002** | 0.3% | 0.8 |
| EDGE | **258** | 0.146 | 13,423 | 1.70 | 0.018 | **0.002** | 0.4% | 62.0 |
| ARROW-Diff | 478 | **-0.098** | **49,865** | **1.78** | **0.069** | 0.001 | **34.2%** | 11.2 |
| *PubMed* | 171 | -0.044 | 12,520 | 2.18 | 0.060 | 0.004 | - | - |
| NetGAN | **150** | -0.021 | 184 | 1.90 | 0.001 | 0.000 | 0.1% | 39.7 |
| Graphite | 918 | **-0.209** | 4 M | 1.31 | 0.341 | 0.001 | 0.3% | 1.3 |
| EDGE | 131 | 0.027 | **2,738** | **2.03** | 0.005 | 0.001 | 0.2% | 92.7 |
| ARROW-Diff | 478 | -0.082 | 44,120 | 1.90 | **0.039** | 0.001 | **42.7%** | 14.4 |

**Baseline Methods**   We use four different graph generation baseline methods, which are designed for training on single graphs to compare against our method: VGAE (Kipf & Welling, 2016b), Graphite (Grover et al., 2019), NetGAN Bojchevski et al. (2018), and EDGE (Chen et al., 2023). To train the baseline methods, we use the recommended hyper-parameters from their papers and code. Depending on the method, node features were used to train VGAE, Graphite, and ARROW-Diff, but were not used for NetGAN and EDGE. The training of NetGAN is performed using their proposed VAL-criterion (Bojchevski et al., 2018) for early stopping on the validation edges from the data split. The models for EDGE were trained for several days on the five datasets. However, only the model on the CiteSeer dataset converges after 2600 epochs. For the other datasets, we consider the models after 5550 (Cora-ML), 250 (Cora), 450 (DBLP), and 250 (PubMed) epochs of training. Additionally, to fit into GPU memory, we decreased the batch size from 4 (training) and 64 (validation) to 2 to train the models on the Cora, DBLP, and PubMed datasets. In the case of VGAE, the method generated over 50 M edges on the Cora, DBLP, and PubMed datasets, which led to an exhaustive metric computation. Thus, in Table 2, we leave out the results on these datasets.

Table 3: Graph Generation performance of GRAN (Liao et al., 2019), GraphRNN (You et al., 2018), Digress (Vignac et al., 2023), EDGE (Chen et al., 2023) and our method ARROW-Diff in the multi-graph setting. Performance is reported using the Maximum Mean Discrepancy (MMD) on three graph statistics, namely Degree, Orbit, and Clustering coefficient.

| Dataset | Method | Degree↓ | Orbit↓ | Clustering↓ | Time/Epoch |
|---------|--------|---------|--------|-------------|------------|
| Community-20 | GRAN | 0.065 | 0.048 | 0.170 | 0.6s |
| | GraphRNN | 0.048 | 0.014 | 0.094 | 1.5s |
| | Digress | **0.025** | 0.008 | **0.009** | 0.5s |
| | EDGE | 0.028 | **0** | 0.931 | 3.2s |
| | ARROW-Diff | 0.105 | 0.075 | 0.237 | **0.05**s |
| CiteSeer-Small | GRAN | 0.018 | 0.015 | 0.014 | 0.7s |
| | GraphRNN | 0.403 | 0.737 | 0.366 | 1.3s |
| | Digress | **0.009** | 0.010 | **0.012** | 0.6s |
| | EDGE | 0.012 | **0** | 0.033 | 4.7s |
| | ARROW-Diff | 0.031 | 0.002 | 0.035 | **0.05**s |

**Evaluation of Generated Graphs**  We use 6 different graph metrics to evaluate the performance of the trained models. Additionally, we report the edge overlap (EO) between the generated graphs and the original graph/LCC. Specifically, we generate 10 graphs per dataset and compute the mean of the metrics to have a better estimate of the performance.

## 5.3 TRAINING GRAPH GENERATION MODELS ON MULTI-GRAPH DATASETS

**Datasets**  In this setting, we use two graph datasets containing undirected graphs: (1) The CiteSeer-Small dataset from You et al. (2018), which consists of 200 ego graphs split into 160/40 for training/testing respectively, with 20% of training split used for validation and with a maximum of 20 nodes per graph; (2) The Community-20 dataset from Martinkus et al. (2022), which consists of 100 random community graphs with 12 to 20 nodes per graph. The graphs in the Community-20 dataset are split into parts of 64/20/16 graphs for training/testing/validation, respectively. The same splits for both datasets were used consistently across all four baseline methods.

**Baseline Methods**  In this setting, we compare ARROW-Diff to four different baseline methods that use multiple graphs for training and testing: GraphRNN (You et al., 2018) and GRAN (Liao et al., 2019), two autoregressive non-diffusion-based approaches, and two diffusion-based models, DiGress (Vignac et al., 2023), which is non-autoregressive, and EDGE (Chen et al., 2023), which is autoregressive. For all baselines, we use the list of hyper-parameters recommended by the authors in their respective papers. These can be found in the supplementary materials.

**Evaluation of Generated Graphs**  To compare ARROW-Diff with methods that use multiple graphs for training, we train one model per graph in the training split as suggested by You et al. (2018). To evaluate the quality of the generated graphs w.r.t. the graphs in the test split, we sample 10 graphs from each of the trained models. Then, we use the $10\times$(number of trained models) generated graphs to evaluate the quality of the samples by calculating the Maximum Mean Discrepancy (MMD) over the degree, orbit, and clustering coefficient between the generated graphs and original graphs. To calculate MMD, we use the Wasserstein distance also known as earth mover's distance (EMD).

## 5.4 RESULTS AND EFFICIENCY

The results pertaining to the first setting are presented in Table 2. Here, our method exhibits a significant improvement across most metrics and outperforms all baselines in terms of the average clustering coefficient. It also shows a higher edge overlap with the original graph across all datasets. The standard deviation of all metrics over the 10 runs is shown in Table 4. Furthermore, the scalability of our approach exceeds all baseline methods designed for large graph generation like NetGAN (Bojchevski et al., 2018) and EDGE (Chen et al., 2023), which is reflected both in terms

of training speed and graph generation time. Our method, ARROW-Diff, demonstrates a substantial decrease in graph generation time even when generating very large graphs such as Cora, PubMed, and DBLP (Table 1), where we can see a decrease of more than 50%. This is shown in column 'Time' in Table 2. As for the training speed, and thanks to the power of the OA-ARDM, our random walk-based diffusion model converges only within 30 minutes, whereas EDGE, the second-best performing method, requires over 4 days on most datasets. Notably, EDGE performs the best in terms of maximum node degree across all datasets. This is due to its ability to steer the graph generation process towards a degree distribution similar to that of the original graphs. In Table 3 we show the results for the second setting, in which we train on datasets consisting of multiple graphs. Here, ARROW-Diff exhibits a comparable performance across the three metrics. However, our approach shows a significant advantage in terms of training speed, almost 10 times faster than Digress (Vignac et al., 2023), the method with the best performance. These results are indicated in Table 3 as time/epoch. It also requires far fewer training iterations with a maximum of 3k epochs across all training graphs compared to 100k epochs for Digress. In the appendix, we provide some visualizations of the generated graphs from ARROW-Diff as well as from all baseline methods in Figure 2 and Figure 3.

## 6 Complexity Analysis

In the following let $N$ denote the number of nodes and $|E|$ the number of edges in a graph, $D$ the random walk length, and $L$ the number of generation steps of ARROW-Diff. In each generation step $l \in [1, L]$, ARROW-Diff first samples $M$ conditional random walks of length $D$ for each start node $n \in V_{\text{start}}$ to compute edge proposals. This has a time complexity of $\mathcal{O}(NMD)$ because $V_{\text{start}} \subseteq V$ and $V_{\text{start}} = V$ in the first step. Next, ARROW-Diff uses a GNN, e.g. a GCN Kipf & Welling (2016a), to compute the probabilities for each edge in the generated graph up to this step, including the set of proposed edges, which requires $\mathcal{O}(|E|)$ operations. The computation of the new start nodes for the next iteration requires $\mathcal{O}(|E|)$ operations with a complexity of $\mathcal{O}(|E|)$ for computing the node degrees of $\hat{G}$ and $\mathcal{O}(N)$ to compute the probabilities and sample the new start nodes. Hence, for $L$ generation steps, ARROW-Diff has a run time of $\mathcal{O}(L(NMD + |E|))$.

## 7 Conclusion

In this paper, we present ARROW-Diff, a novel graph generation approach based on random walk diffusion. Our method demonstrates scalability to very large graphs, surpassing the capability of existing baselines. This scalability is achieved through the efficient training and sampling of the OA-ARDM and the generation time of ARROW-Diff, which shows a significant decrease compared to all baselines. It is worth mentioning that we also implemented the D3PM discrete diffusion process on the level of random walks. However, this caused a notable increase in training and generation time. Moreover, our approach is directly applicable to both directed and undirected graphs. To demonstrate the performance of our approach, we compare ARROW-Diff in two different experimental settings to multiple baseline methods. ARROW-Diff outperforms most of these methods on multiple graph statistics, or at least competes with them. Nevertheless, one limitation of our approach is that it can only generate graphs with the same number of nodes as the original graph, due to the behavior of the discrete, autoregressive diffusion model. Potential future work could focus on a better adaptation of ARROW-Diff for learning on multiple graphs.

## Reproducibility Statement

In the supplementary materials we provide the full implementation of ARROW-Diff, along with a README file of how to run our code. We also provide configuration files containing all parameters used for training and evaluation of our method and all baselines.

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

## A   STANDARD DEVIATIONS OF RESULTS ON THE SINGLE GRAPHS

Table 4: Standard deviation for each metric shown in Table 2.

| *Dataset* Methods | Max. degree | Assort-ativity | Triangle Count | Power law exp. | Avg. cl. coeff. | Global cl. coeff. | Edge Overlap | Time [s] |
|---|---|---|---|---|---|---|---|---|
| Cora-ML | 246 | -0.077 | 5,247 | 1.77 | 0.278 | 0.004 | - | - |
| NetGAN | 11 | 0.004 | 19 | 0.00 | 0.001 | 0.000 | 0.2% | 1.0 |
| VGAE | 24 | 0.003 | 2 M | 0.11 | 0.001 | 0.000 | 0.6% | 0.0 |
| Graphite | 14 | 0.015 | 1,411 | 0.02 | 0.007 | 0.001 | 0.1% | 0.0 |
| EDGE | 1 | 0.010 | 125 | 0.00 | 0.003 | 0.000 | 0.1% | 3.7 |
| ARROW-Diff | 9 | 0.002 | 320 | 0.02 | 0.007 | 0.000 | 1.5% | 0.0 |
| Cora | 297 | -0.049 | 48,279 | 1.69 | 0.267 | 0.007 | - | - |
| NetGAN | 13 | 0.003 | 13 | 0.00 | 0.000 | 0.000 | 0.0% | 0.9 |
| Graphite | 79 | 0.005 | 116,224 | 0.00 | 0.004 | 0.000 | 0.0% | 0.0 |
| EDGE | 3 | 0.016 | 1,111 | 0.00 | 0.001 | 0.000 | 0.0% | 0.8 |
| ARROW-Diff | 15 | 0.003 | 2,040 | 0.00 | 0.002 | 0.000 | 0.7% | 0.7 |
| CiteSeer | 85 | -0.165 | 771 | 2.23 | 0.153 | 0.007 | - | - |
| NetGAN | 8 | 0.019 | 6 | 0.01 | 0.002 | 0.000 | 0.2% | 0.2 |
| VGAE | 16 | 0.004 | 337,244 | 0.11 | 0.001 | 0.000 | 0.8% | 0.1 |
| Graphite | 6 | 0.022 | 419 | 0.02 | 0.010 | 0.001 | 0.1% | 0.0 |
| EDGE | 0 | 0.011 | 27 | 0.01 | 0.007 | 0.000 | 0.2% | 3.3 |
| ARROW-Diff | 5 | 0.007 | 69 | 0.03 | 0.008 | 0.000 | 1.4% | 0.0 |
| DBLP | 339 | -0.018 | 36,645 | 1.76 | 0.145 | 0.004 | - | - |
| NetGAN | 10 | 0.005 | 69 | 0.00 | 0.000 | 0.000 | 0.0% | 0.4 |
| Graphite | 74 | 0.004 | 69,026 | 0.00 | 0.003 | 0.000 | 0.0% | 0.0 |
| EDGE | 2 | 0.033 | 1,675 | 0.00 | 0.002 | 0.000 | 0.0% | 0.5 |
| ARROW-Diff | 26 | 0.002 | 2,202 | 0.01 | 0.002 | 0.000 | 0.8% | 0.7 |
| PubMed | 171 | -0.044 | 12,520 | 2.18 | 0.060 | 0.004 | - | - |
| NetGAN | 14 | 0.004 | 9 | 0.00 | 0.000 | 0.000 | 0.0% | 1.8 |
| Graphite | 70 | 0.005 | 244,816 | 0.00 | 0.004 | 0.000 | 0.0% | 0.0 |
| EDGE | 4 | 0.038 | 741 | 0.00 | 0.001 | 0.000 | 0.0% | 0.5 |
| ARROW-Diff | 11 | 0.003 | 1,454 | 0.00 | 0.001 | 0.000 | 0.8% | 1.1 |

# B VISUALIZATION OF GENERATED GRAPHS

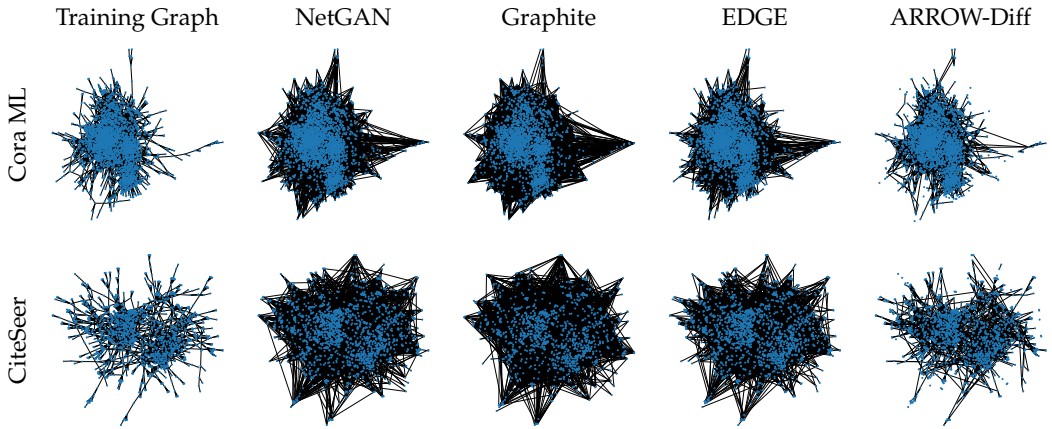

Figure 2: Visualization of the training graphs and generated graphs for Cora-ML and CiteSeer from NetGAN, Graphite, EDGE, and ARROW-Diff using the trained models from Section 5.2.

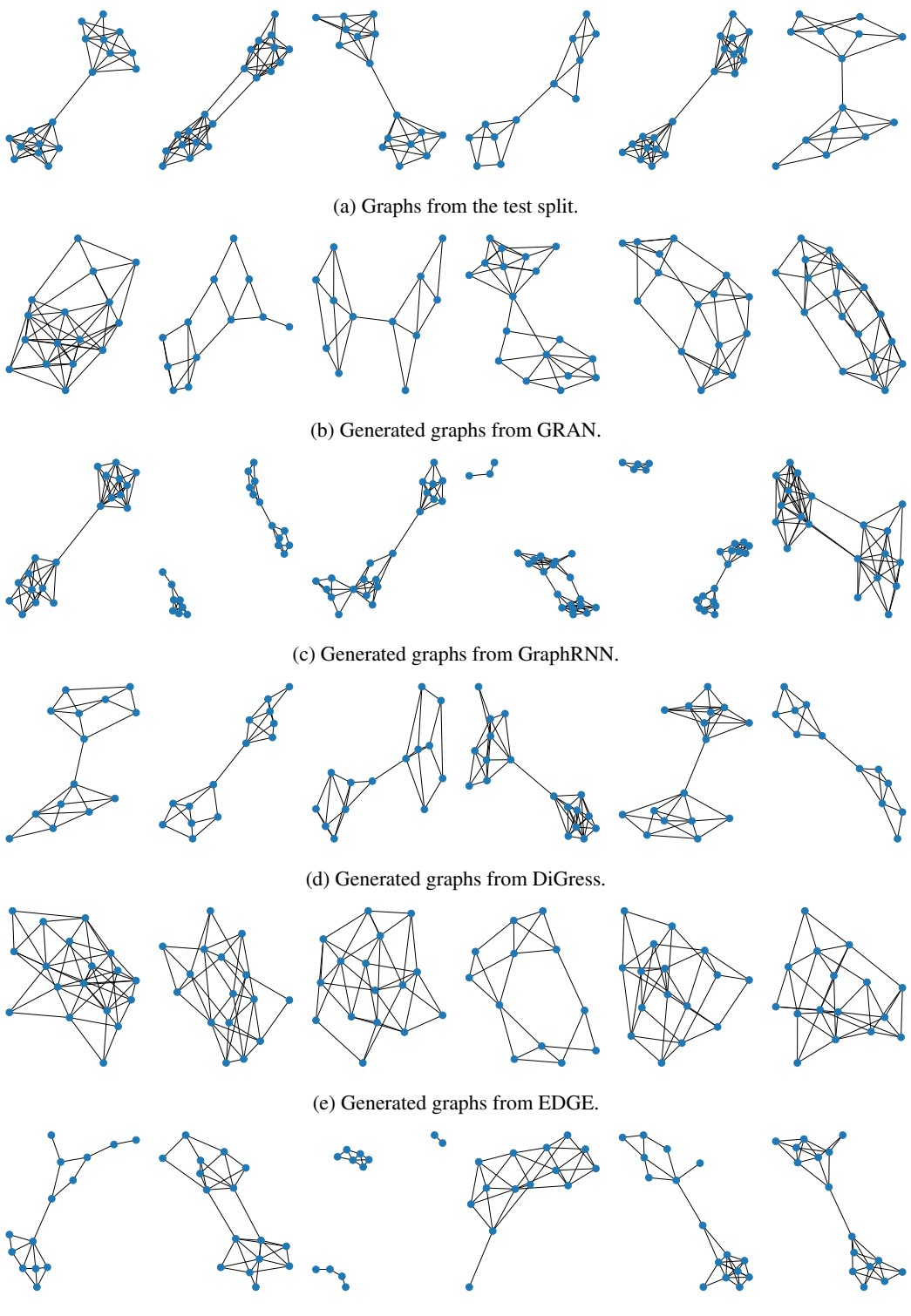

(a) Graphs from the test split.

(b) Generated graphs from GRAN.

(c) Generated graphs from GraphRNN.

(d) Generated graphs from DiGress.

(e) Generated graphs from EDGE.

(f) Generated graphs from ARROW-Diff.

Figure 3: Visualization of six generated graphs from all baseline methods and ARROW-Diff trained on the Community-20 dataset (Section 5.3).

