# OpenReview forum: "Random Walk Diffusion For Graph Generation"
_ICLR.cc/2024/Conference — Submitted to ICLR 2024_

### Official Review · Reviewer_sVdq · 2023-10-28

**Soundness:** 2 fair
**Presentation:** 2 fair
**Contribution:** 3 good
**Rating:** 3
**Confidence:** 5

**Summary:**

The paper proposes a novel graph generative model for modeling very large graphs. The main idea is to sample random walks, which generate candidate edges for the current graph t, then a GNN will run on such a noisy graph and prune out some candidate edges. Such iteration is repeated until the graph is finalized. The key intuition of the paper is to use the sparsity of the graph obtained from the random walks, which can be handled properly by the GNN. To obtain those random walks, the paper proposes to use the BERT-like model to sample node id sequences.

**Strengths:**

The idea is pretty novel and it's interesting. The model tackles the dense graph generation problem elegantly by using two sets of model (random walk and GNN refinement.

**Weaknesses:**

1. Notation is unlcear. For example, it is not clear how equation 2 is computed, and what's the model like. When the authors say the random walks are sampled by the BERT-like model, it's not clear whether it's permutation-invariant or not since the BERT model doesn't come with the causal mask but with positional embedding.

2. it's not clear how the V_start is reset using node degree information.

3. In my understanding, the random walk sampling doesn't utilize any structural information of the (previous) generated graph. It must take the node feature information as input in order to work well. In this case, the proposed method can only generate graphs conditioning on node features. Since node features are not considered by NetGAN and EDGE, I suggest the author should highlight the difference.

4. Based on my comment 3, the author should also add graphite as one of the baselines.

5. I believe the Edge Overlap is the lower the better. If a model works well with high EO, it is simply because it memorizes the original graph structure. I don't think the comparison against the baseline is totally fair -- Arrow-diff has 50% EO while the others have ~1% EO

6. Is the result reported in EDGE and NetGAN trained with node features? if so, can you provide details about how the model is modified to do so?

7. Time complexity analysis is missing. It should be explicitly analyzed to demonstrate why the method is so efficient, this involves showing the detail of the used models (for random walk and GNN).

**Questions:**

See weakness, I'd like to change my rating if the concerns are addressed.

---

> ### Author Response · Authors · 2023-11-15
>
> We thank the reviewer for the thorough feedback. We read the comments carefully and will consider them in the revised version. Following are our answers to the specific questions.
>
> **Question 1:**
>
> Equation 2 gives the log-likelihood of time step $t$ for the Order Agnostic Autoregressive Diffusion Model [1] (OA-ARDM). It is derived from Equation 1 (Section 2 “Background” in our paper). The details of this derivation can be found in Hoogeboom et al. [1] in paragraph “Order Agnostic ARDMs” in Section 3 of their paper. We are happy to provide more information if needed.
>
> About our model (see Section 5.1): We use a U-Net architecture similar to Ho et al. [2] with one ResNet block and two levels for the down- and up-sampling processes. For the graph generation process, we train a 2-layer GCN to classify edges into valid/invalid ones based on perturbed versions of the input graph. We are ready to provide more clarification if needed.
>
> Please also check out our code in the supplementary material. It is well documented and gives the correct dimensionalities of tensors and should help to understand e.g. the computation of Equation 2.
>
> Could you please clarify whether you mean “permutation-invariant” or “order-agnostic” here? More specifically, our model is order-agnostic and can generate random walks in any order. This is ensured by the order-agnostic training paradigm, in which for each sampled permutation of a random walk $D-t+1$ tokens are masked out and then predicted. This is what Hoogeboom et al. [1] refer to as BERT-like training.
>
> **Question 2:**
>
> The new start nodes $V_\text{start}$ in line 11 of Algorithm 2 are sampled from $V$ independently from each other using a Bernoulli distribution, where sampling a ‘1’ corresponds to including a node $n$ to $V_\text{start}$ and sampling a ‘0’ corresponds to not including $n$ to $V_\text{start}$. Each node $n \in V$ has its own probability $p(n)$, which is used to sample from the Bernoulli distribution and is computed as follows:
>
> 1. Compute the element-wise (or node-wise) differences in node degrees between the original graph $G$ and the current (at generation step $l$) generated graph $\hat{G}$: $\quad\boldsymbol{d}_\Delta := \boldsymbol{d}_G - \boldsymbol{d}\_\hat{G}$
>
> 2. Compute the element-wise ReLU of $\boldsymbol{d}_\Delta$ : $\quad \boldsymbol{d} := \max(0, \boldsymbol{d}\_\Delta)$
>
> 3. Compute the element-wise (node-wise) probabilities $p(n)$ for sampling node $n$ into $V_\text{start}$ by dividing each entry in $\boldsymbol{d}$ by the maximum entry in $\boldsymbol{d}$: $\quad p(n) = d_n / \max(\boldsymbol{d})$
>
> Note: The values of $p(n)$ do not sum to 1, i.e. $\sum_{n \in V} p(n) \neq 1$, because each value $p(n)$ gives the probability whether to include node $n$ into $V_\text{start}$ or not, i.e. $p(n)$ gives the probability of sampling a ‘1’ from a Bernoulli distribution and $1 - p(n)$ gives the probability of sampling a ‘0’ from a Bernoulli distribution.
>
> We thank the reviewer for bringing this important point to our attention. This was not properly described in the paper. We will update the paper accordingly.
>
> **Question 3:**
>
> There is a typo in our Algorithm 2 in reflecting this. Thank you for pointing this out. The GNN of our iterative procedure does indeed use the structure (edges) of the generated graph from the previous step $l-1$. We will change it in the paper accordingly.
>
> Node features are not considered by NetGAN and EDGE. We explain more about that in our answer to your related question 6.
>
> **Question 4:**
>
> We plan to add the Graphite baseline model to our evaluation in the revised paper.
>
> **Question 5:**
>
> "I believe the Edge Overlap is the lower the better." - This strongly depends on the type (and desired properties) of the graph to generate and thus is a matter of interpretation. There is for sure a trade-off between the edge overlap and the novelty of a generated graph. However, this trade-off is not yet well explored in the literature. Nonetheless, given the performance of our method on the rest of the metrics, we think we are not totally overfitting but rather seeing the novelty of the generated graphs.
> Here, it is worth mentioning that although we report high EO on the Cora-ML dataset, the performance on other metrics does not reflect better quality (here compared to EDGE). At the same time, we report a lower EO on DBLP but better performance on the rest of the metrics. This shows that the EO and performance are not necessarily correlated.
>
>
> ---
> [1] Emiel Hoogeboom, Alexey A. Gritsenko, Jasmijn Bastings, Ben Poole, Rianne van den Berg, and Tim Salimans. Autoregressive diffusion models. In International Conference on Learning Representations, 2022.
>
> [2] Jonathan Ho, Ajay Jain, and Pieter Abbeel. Denoising diffusion probabilistic models. Advances in neural information processing systems, 33:6840–6851, 2020.

---

> ### Author Response · Authors · 2023-11-15
>
> **Question 6:**
>
> NetGAN does not use node features. The method samples random walks from the training graph and trains an LSTM-based Generative Adversarial Network (GAN) for random walk generation. EDGE does not use the node features of the training graph as well.
> We will make sure to highlight this in the paper.
>
> The code of all baselines in the paper was not modified in a way that would change the behavior and methodology of the baselines. Nevertheless, the data loading parts in the code of the baselines were adapted for loading the datasets in our paper.
>
> **Question 7:**
>
> We will provide a comprehensive time complexity analysis as part of the revised paper.

---

### Official Review · Reviewer_ynZg · 2023-10-29

**Soundness:** 4 excellent
**Presentation:** 3 good
**Contribution:** 3 good
**Rating:** 6
**Confidence:** 4

**Summary:**

The paper proposes a novel graph generation approach by designing the diffusion models on random walks. It uses an order agnostic autoregressive diffusion model to sample random walks from a given graph, and then uses a graph neural network to predict the edge set in the final graph. The paper claims that this approach can generate high-quality graphs that are similar to real-world graphs and can scale to very large graphs efficiently.

**Strengths:**

1.	The paper introduces a new perspective of applying diffusion models to graph generation by developing a diffusion process on the random walks.
2.	The paper demonstrates the scalability and flexibility of the proposed approach, which can handle both directed and undirected graphs, and both single-graph and multi-graph datasets.
3.	The paper provides extensive experiments and comparisons with several baselines on various graph metrics, showing the superiority of the proposed approach in terms of quality and speed.

**Weaknesses:**

1.	It is better for the authors to provide qualitative examples or visualizations of the generated graphs, which would help to illustrate the effectiveness and diversity of the proposed approach.
2.	Based on my understanding, the effectiveness and efficiency of the proposed methods are both sensitive to the parameter L. The authors should provide the necessary analysis on the selection of this parameter. Intuitively, when using the large L, the inference time would increase linearly.
3.	In Table 2, the performance of the proposed method seems to vary a lot on graphs with different sparsity. It is better for the authors to provide more empirical analysis.

**Questions:**

Please provide several visualizations of the generated graphs.
How does the parameter L influence the performance?
Is there a significant correlation between the performance of the proposed method and the sparsity of graphs? What is the relationship?

---

> ### Author Response · Authors · 2023-11-22
>
> We thank the reviewer for their feedback.
>
> **Weakness 1:**
> We are providing a visualization of the generated graphs from our method as well as all baselines in the appendix in the revised version of our paper.
>
> **Weakness 3:**
>
> Our calculations show that the adjacency matrices of the graphs in the single graph datasets have a density percentage between 0.04% - 0.23%. However, our results (in terms of the metrics) do not seem to vary a lot. For example, the sparsity of Cora is 0.04% and the sparsity of CiteSeer is 0.21%, but the results on these two datasets do not show a lot of variation.

---

### Official Review · Reviewer_Yei3 · 2023-10-30

**Soundness:** 3 good
**Presentation:** 3 good
**Contribution:** 2 fair
**Rating:** 3
**Confidence:** 4

**Summary:**

The paper introduces a generative model to generate edges of a graph. It is based on Order Agnostic AutoRegressive Diffusion Models (OA-ARDM) that capture dependencies of the input and are robust to the order in which intermediate samples are generated. The approach is an adaptation of OA-ARDM (Hoogeboom et al., 2022) to graphs where each sample at some diffusion step t is a random path of size t in the graph. The model is trained by masking those paths and predicting them.

**Strengths:**

In terms of methodology, the proposed approach is a straigthforward adaptation of (Hoogeboom et al., 2022) to sequences made of random walks in a graph. The paper is well-written and the method seems to obtain train faster than baselines assuming that the number of iterations (i.e. sampled paths/random walks) is relatively small and their length is short.

**Weaknesses:**

1) At inference time, the method relies on generating a sufficient number of paths with appropriate length. Assuming that the training graphs are densely connected (i.e. a large number of pairs of nodes are connected by an edge), the number of edges in the graph is quadratic in the number of nodes, and the number of paths increases significantly as their length increases. It is unclear how the proposed method is robust and representative of all the possible paths in the graphs. In particular, the method would miss a lot of edges if it does not sample enough random walks.

Similarly, during training, the method only considers local information in the graph (i.e. random edge sequences) and not its global structure. Therefore, for large dense graphs, the number of edges becomes quadratic in the number of nodes and it might be difficult to sample enough random walks to be representative of the connectivity in the graph.

2) In terms of evaluation, the purpose of the task described in Section 5.2 (graph generation from a single-graph dataset) is not clear to me. The purpose of graph generation is usually to generate multiple diverse graphs that follow the same distribution as the graphs forming a dataset. If the training set contains a single graph, then it is not clear what the distribution of the training set should be other than a singleton. A naive baseline for the task would be an autoencoder whose decoder always returns the same graph as the training graph. It also seems that the proposed method corresponds to a (denoising) autoencoder that simply tries to reconstruct the edges of a single graph in the setup of Section 5.2. This would explain why the proposed method outperforms the baselines in this setup.

The evaluation with the reported baselines then seems unfair since many baselines are trained to promote novelty and non-uniqueness (i.e., generating graphs that are not in the training set, and diverse).
The scores reported in Table 2 seem to promote overfitting over a single sample/graph and does not really reflect the generative power of the different methods.

3) The experiments in Section 5.3 and Table 3 are not convincing either. Digress and EDGE seem competitive with respect to all the evaluation metrics except training time (which may be reduced with more computing power). In generative models, an interesting metric is the inference time to generate new samples. The novelty and uniqueness score are not reported either.

4) The number of nodes of the generated graph is the same as the number of nodes of the training graph.

**Questions:**

1) What are the novelty and uniqueness scores of the different methods in Table 3?

2) How important is initialization for the generation process? Assuming that the number of steps L and the number of initial sampled random walks M are both large, do different sets of random walks tend to return the same value of p(n) in Step 11 of Algorithm 2?

3) The main purpose of using multiple GNN steps at inference time is to evaluate the degrees of nodes. Do you have ablation studies to see the impact of M and L for training? If M becomes large, then the training takes longer but does it improve generation scores? Same question for the length of the paths.

4) How does the method deal with permutation invariance? Assuming that the dataset contains multiple graphs that are all isomorphic (e.g. the graph with the edges (1, 2) and (2, 3) is isomorphic with the graph containing (3, 1) and (1, 2), or (1, 3) and (3,2)), will the values of p(n) converge to some canonical representation or will p(n) = 1/3 for all n in this case?

5) Same question for when the graph contains different nonisomorphic graphs, how does the method deal with permutation of nodes? How does the GNN differentiate the node IDs between different graphs?


Minor detail. The paper should mention that m in step 3 of Algorithm 1 is a Boolean mask.

---

> ### Author Response · Authors · 2023-11-17
>
> We thank the reviewer for the thorough feedback. We read the comments carefully and will consider them in the revised version. Please find the answers to the specific questions below.
>
> **Weakness 1:**
>
> During training, the number of sampled random walks for each trained model (whether in the single or multiple-graph setting) was not fixed a priori, rather we sample a ‘batch_size’ number of random walks in every epoch and use early stopping as criterion to stop the training. During sampling, given that our iterative sampling process is guided by the change in node degree distribution, this means that the densely connected nodes will have higher probability to be chosen as start nodes to sample from for the next iteration. However, the goal of generating new graphs is not to duplicate the edges of the original graph, but rather to create graphs with a similar structure that are captured by the wide range of evaluation metrics.
>
> The robustness of our evaluation metrics is ensured by repeating the sampling process for 10 times. We thank you for pointing this out. We will include the standard deviations of all metrics across these 10 runs in the Appendix.
>
> It is worth mentioning that the citation graphs that we use are the largest and most dense graphs used in the graph generation literature so far [1, 2].
> As for capturing local vs. global information: Similar to NetGAN we also use the random walk sampling strategy described in Grover & Leskovec [3], which would ensure that we also capture global structural information from the input graph.
>
> **Weakness 2:**
>
> Since obtaining a set of graphs that all come from the same distribution is hard for real-world data, the task of generating new graphs using a single training graph becomes vital, e.g. to classify real vs. fake graphs like social networks [4]. Here the task is reduced to learning the specific structural information of a graph (evaluated in terms of metrics like clustering coefficient, power-law exponent, etc.). To do this we propose to learn this structural information by modeling random walks sampled from the training graph and then using the walks that are generated via a denoising diffusion model to build the final graph in an iterative procedure. Other methods that use a single graph for learning the generative process include NetGAN and EDGE.
>
> Given the performance of our method on the rest of the metrics, we think we are not totally overfitting, but rather seeing the novelty of the generated graphs. Here, it is worth mentioning that although we report high EO on the Cora-ML dataset, the performance on other metrics do not reflect better quality (here compared to EDGE). At the same time, we report a lower EO on DBLP but better performance on the rest of the metrics. This shows that the EO and performance are not necessarily correlated.
>
> **Weakness 4:**
>
> Indeed we have mentioned this as a limitation of our method in Section 6 of our paper.
>
> **Question 2:**
>
> Assuming we want to generate two graphs using the same training graph for the diffusion model and GNN: If for two sets of random walks $A$ and $B$ the edges inside $A$ lead to the nodes of the generated graph $G_A$ having the same node degrees as for a generated graph $G_B$, the values of $p(n)$ would be the same for every node in both graphs for different generation processes.
> If in two different (following) steps of the same generation process the edges in the proposed random walks from the diffusion model do not lead to new edges (e.g. if all are marked as invalid by the GNN), then the values of $p(n)$ would be the same in these two iterations.
> Both cases remain true whether $L$ and $M$ are small or large.
>
> ---
>
> [1] Aleksandar Bojchevski, Oleksandr Shchur, Daniel Zügner, and Stephan Günnemann. NetGAN: Generating graphs via random walks. In Jennifer Dy and Andreas Krause (eds.), Proceedings of the 35th International Conference on Machine Learning, volume 80 of Proceedings of Machine Learning Research, pp. 610–619. PMLR, 10–15 Jul 2018.
>
> [2] Xiaohui Chen, Jiaxing He, Xu Han, and Li-Ping Liu. Efficient and degree-guided graph generation via discrete diffusion modeling. arXiv preprint arXiv:2305.04111, 2023.
>
> [3] Grover, A. and Leskovec, J. node2vec: Scalable feature learning for networks. In Proceedings of the SIGKDD international conference on Knowledge discovery and data mining, pp. 855–864, 2016.
>
> [4] Chakrabarti, D. and Faloutsos, C. Graph mining: Laws, generators, and algorithms. Computing Surveys (CSUR), 38(1):2, 2006.

---

> ### Author Response · Authors · 2023-11-17
>
> **Weakness 3:**
> The reason behind the lower performance in the multiple-graph setting is due to the fact that we train one model for each graph in the training dataset as was suggested by You et al. [2]. Thus, these models do not share parameters. For a fair comparison, we do not report the sampling time but rather focus on the advantage of our method on training time in the multi-graph setting. This is because, in our current training strategy, we need to sample multiple graphs from each trained model, it is something we would like to improve in our future work. However, the superiority of our method in sampling time is reflected in the first setting of our experiments (on single large graphs).
>
> **Question 3:**
>
> The purpose of using GNN during inference is to check for plausible edges in the generated graph based on what it had learned from the original graph.
> $L$ and $M$ are only used in the graph generation process (Algorithm 2). They are not used for training of the diffusion model or the GNN.
> We do not sample multiple random walks per node for each training loop of the diffusion model. We control the training time (iterations) using different numbers of epochs with early stopping.
> Regarding the lengths of the random walks: We fixed $D = 16$ for the graph generation experiments on single graph datasets following the insights from Bojchevski et al. [1] and to ensure a fair comparison. In the multi-graph setting we experimented with $D \in \{8, 12\}$, where $D = 12$ led to the best results.
>
> **Questions 4 and 5:**
>
> In our multi-graph setting we train one model (diffusion model and GNN) for each training graph in the multi-graph training set as suggested by You et al. [2], i.e. when generating a new graph, $p(n)$ is independent across the models that were trained on the different graphs. Thus, also the node IDs of different graphs do not affect each other.
> The diffusion model only predicts the node indices in the random walks, our model is not affected by the ordering of nodes (which is random, but fixed before training) and is thus permutation invariant.
>
> ---
>
> [1] Aleksandar Bojchevski, Oleksandr Shchur, Daniel Zügner, and Stephan Günnemann. NetGAN: Generating graphs via random walks. In Jennifer Dy and Andreas Krause (eds.), Proceedings of the 35th International Conference on Machine Learning, volume 80 of Proceedings of Machine Learning Research, pp. 610–619. PMLR, 10–15 Jul 2018.
>
> [2] Jiaxuan You, Rex Ying, Xiang Ren, William Hamilton, and Jure Leskovec. GraphRNN: Generating realistic graphs with deep auto-regressive models. In Jennifer Dy and Andreas Krause (eds.), Proceedings of the 35th International Conference on Machine Learning, volume 80 of Proceedings of Machine Learning Research, pp. 5708–5717. PMLR, 10–15 Jul 2018.

---

### Official Review · Reviewer_zEev · 2023-10-31

**Soundness:** 2 fair
**Presentation:** 3 good
**Contribution:** 2 fair
**Rating:** 6
**Confidence:** 4

**Summary:**

Existing diffusion-based graph generation models are designed for generating small graphs and suffer from scaling to large-scale graphs. To scale diffusion-based models to generate large graphs, this work proposes ARROW-Diff, Auto Regressive RandOm Walk Diffusion, for graph generation based on random walk diffusion. ARROW-Diff leverages order-agnostic Autoregressive Diffusion Models (OA-ARDMs) to sample random walks from the training graphs, and further encompasses an iterative procedure that generates the final graph by utilizing a Graph Neural Network (GNN) model to filter out invalid edges from sampled random walks. Experiments on both a single, large-scale graph setting and the setting of multiple, small graphs demonstrate the efficiency and scalability of ARROW-Diff.

**Strengths:**

1. This work scales the diffusion-based approaches for large-scale graph generation in a random walk diffusion fashion. Inspired by node2vec, ARROW-Diff utilizes OA-ARDMs to sample random walks from training graphs, and then leverages an iterative procedure to construct the final graph based on an edge classification task directed by node degrees from the sampled random walks. The idea is natural and the ARROW-Diff framework is sound.

2. The authors did a great job of introducing the background and related work. The illustration of the ARROW-Diff procedure, especially Figure 1, is clear and easy to understand.

3. Regarding experiments on the single, large-scale graph setting, ARROW-Diff outperforms the existing graph generation baselines by a certain margin in terms of the runtime, and also generates graphs with higher quality than baselines in terms of relatively more complex graph metrics (e.g., triangle count, edge overlap).

**Weaknesses:**

1. The main contribution of this work is leveraging OA-ARDMs to generate large-scale graphs efficiently in a random walk diffusion fashion. Although ARROW-Diff demonstrates its efficiency and scalability empirically, the contribution in terms of the idea is not very novel, and the authors did not give any theoretical justification (e.g., time complexity analysis) for why ARROW-Diff is more efficient than existing graph generation methods.

2. Although ARROW-Diff is able to generate graphs much more efficiently than baselines in the setting of multiple, small graphs, the quality of generated graphs by ARROW-Diff does not outperform baselines in terms of all three metrics.

3. (Minor) I did not find any appendix for this work in the supplementary materials. It would be great if the authors could further describe the details of the implementation of ARROW-Diff and the experiments.

**Questions:**

1. Regarding the first point in the Weaknesses, I wonder if the authors could provide a more theoretical analysis of the complexity of ARROW-Diff.

2. Regarding the second point in the Weaknesses, I wonder if the authors could discuss further whether ARROW-Diff is able to generate high-quality graphs in the single, large-scale graph setting while not outperforming baselines in the settings of multiple, small graphs.

3. Recently, there is another paper related to diffusion-based graph generation [1]. With empirical analysis revealing that permutation-invariant diffusion models are harder to learn than their non-permutation-invariant counterparts, [1] proposes a non-permutation-invariant diffusion model for graph generation. I wonder if the authors could discuss about permutation-invariant vs non-permutation-invariant models for the graph generation.

[1] Yan, Q., Liang, Z., Song, Y., Liao, R., \& Wang, L. (2023). Swingnn: Rethinking permutation invariance in diffusion models for graph generation. arXiv preprint arXiv:2307.01646.

---

> ### Author Response · Authors · 2023-11-15
>
> We thank the reviewer for the thorough feedback. We read the comments carefully and will consider them in the revised version. Please find the answers to the specific questions below.
>
> **Weakness 3:**
>
> The details of our method are described in Algorithm 2, and in Section 5.1 “ARROW-Diff Model Training And Sampling”. The details of all experiments are described in Sections 5.2 and 5.3. The corresponding hyperparameters for our models and the baselines can be found in the supplementary material. Please let us know what kind of further details you would like to know.
>
> **Question 1 (and Weakness 1):**
>
> We will provide a comprehensive theoretical time complexity analysis in the updated version of the manuscript.
>
> **Question 2 (and Weakness 2):**
>
> The reason behind the lower performance in the multiple-graph setting is due to the fact that we train one model for each graph in the training dataset as was suggested by You et al. [1]. Thus, these models do not share parameters. Afterwards we sample 10 graphs from each trained model for the evaluation process. This is something we would like to explore in depth in the future. Nonetheless, we wanted to highlight the training speed of our model compared to state-of-the-art multi-graph generation methods.
>
> **Question 3:**
>
> Many existing methods for graph generation try to learn the underlying structure of graphs by modeling the adjacency matrix. Of course,  a process for handling permutation invariance is vital, but node ordering schemes that were suggested in previous methods to handle this problem become really challenging for very large graphs as is the case for GraphRNN [1] and GRAN [2]. In our proposed approach we instead learn the underlying structure of graphs by modeling the context of nodes in random walks by leveraging the order-agnostic autoregressive diffusion training. As the diffusion model only predicts the node indices in the random walks, our model is not affected by the ordering of nodes (which is random, but fixed before training) and is thus permutation invariant.
>
> ---
>
> [1] Jiaxuan You, Rex Ying, Xiang Ren, William Hamilton, and Jure Leskovec. GraphRNN: Generating realistic graphs with deep auto-regressive models. In Jennifer Dy and Andreas Krause (eds.), Proceedings of the 35th International Conference on Machine Learning, volume 80 of Proceedings of Machine Learning Research, pp. 5708–5717. PMLR, 10–15 Jul 2018.
>
> [2] Renjie Liao, Yujia Li, Yang Song, Shenlong Wang, Will Hamilton, David K Duvenaud, Raquel Urtasun, and Richard Zemel. Efficient graph generation with graph recurrent attention networks. Advances in neural information processing systems, 32, 2019.

---

### Author Response · Authors · 2023-11-23
**Updated version of the paper**

We thank all reviewers for their feedback and we have updated our paper accordingly to include:
- Graphite as another baseline in the single graph generation setting.
- A Complexity Analysis section (Section 6).
- A table showing the standard deviation of our results on the task of generating single large graphs (Appendix).
- Visualizations of the generated graphs for single and multi-graph setting from ARROW-Diff as well as from all baselines (Appendix). In the visualization of the single graph setting we only chose the methods for which we had results for all datasets in Table 2. Thus, we left out VGAE in this case.
- An updated version of Algorithm 2 to clarify and incorporate the comments from the reviewers.
- Updated results of VGAE in Table 2 to use a higher cutoff threshold of 0.95 on the probabilities of edges (instead of 0.5 before). This was also done in Graphite for fair comparision. The reason behind this increased threshold is to reduce the number of edges in the final generated graph, which is otherwise having tens of millions of edges for both methods. This was also necessary in order to avoid exhaustive metric computation.

---

### Meta-Review · Area_Chair_DRhd · 2023-12-05

**Metareview:**

The paper introduces ARROW-Diff, a graph generation model that scales to large-scale graphs using Auto Regressive RandOm Walk Diffusion. It combines Order-Agnostic Autoregressive Diffusion Models (OA-ARDMs) to sample random walks from training graphs and a Graph Neural Network (GNN) to iteratively filter out invalid edges, creating more accurate and efficient large graph representations. This approach effectively leverages the sparsity of random walk-generated graphs, allowing for scalable graph generation.

While the proposed ARROW-Diff model shows promising results, the reviewers have identified several weaknesses that need to be addressed:

1. This work primarily utilizes OA-ARDMs for efficient, large-scale graph generation through random walk diffusion; however, it lacks novelty and theoretical justification for its efficiency compared to existing methods.
2. The experimental setting requires further clarification.
3. The presentation needs improvements.

Based on these weaknesses, we recommend rejecting this paper. We hope this feedback helps the authors improve their paper.

**Justification For Why Not Higher Score:**

Two reviewers believed the paper should be rejected, while the other two did not champion the paper during the discussion phase.

**Justification For Why Not Lower Score:**

N/A

---

### Decision · Program_Chairs · 2024-01-16

Reject